# From colon wall to tumor niche: Unraveling the microbiome's role in colorectal cancer progression

Gissel García Menéndez[1], Liubov Sichel[2‡], Maria del Consuelo López[1‡], Yasel Hernández[1‡], Ernesto Arteaga[1‡], Marisol Rodríguez[1‡], Vilma Fleites[3‡], Lipsy Teresa Fernández[4‡], Raúl De Jesus Cano[5] *

1 Pathology Department, Clinical Hospital Hermanos Ameijeiras, Centro Habana, La Habana, Cuba, 2 Stellar Biotics, LLC, Rockleigh, New Jersey, United States of America, 3 Oncology Department Clinical Hospital Hermanos Ameijeiras, Centro Habana, La Habana, Cuba, 4 Surgery Department Clinical Hospital Hermanos Ameijeiras, Centro Habana, La Habana, Cuba, 5 Biological Sciences Department, California Polytechnic State University, San Luis Obispo, CA, United States of America

☯ These authors contributed equally to this work.
‡ These authors also contributed equally to this work.
* rcano@calpoly.edu

**Data Availability Statement:** The SRA sequences have been deposited in GenBank with the BioProjectID PRJNA 1150116. The accession

## Abstract

Colorectal cancer (CRC) is influenced by perturbations in the colonic microbiota, characterized by an imbalance favoring pathogenic bacteria over beneficial ones. This dysbiosis contributes to CRC initiation and progression through mechanisms such as carcinogenic metabolite production, inflammation induction, DNA damage, and oncogenic signaling activation. Understanding the role of external factors in shaping the colonic microbiota is crucial for mitigating CRC progression. This study aims to elucidate the gut microbiome's role in CRC progression by analyzing paired tumor and mucosal tissue samples obtained from the colon walls of 17 patients. Through sequencing of the V3-V4 region of the 16S rRNA gene, we characterized the tumor microbiome and assessed its association with clinical variables. Our findings revealed a significant reduction in alpha diversity within tumor samples compared to paired colon biopsy samples, indicating a less diverse microbial environment within the tumor microenvironment. While both tissues exhibited dominance of similar bacterial phyla, their relative abundances varied, suggesting potential colon-specific effects. Fusobacteriota enrichment, notably in the right colon, may be linked to MLH1 deficiency. Taxonomy analysis identified diverse bacterial genera, with some primarily associated with the colon wall and others unique to this region. Conversely, several genera were exclusively expressed in tumor tissue. Functional biomarker analysis identified three key genes with differential abundance between tumor microenvironment and colon tissue, indicating distinct metabolic activities. Functional biomarker analysis revealed three key genes with differential abundance: K11076 (putrescine transport system) and K10535 (nitrification) were enriched in the tumor microenvironment, while K11329 (SasA-RpaAB circadian timing mediator) dominated colon tissue. Metabolic pathway analysis linked seven metabolic pathways to the microbiome. Collectively,

numbers for each of the SRA files described in the paper are included under the section of "Microbiome Analysis.

**Funding:** The author(s) received no specific funding for this work.

**Competing interests:** The authors have declared that no competing interests exist.

these findings highlight significant gut microbiome alterations in CRC and strongly suggest that long-term dysbiosis profoundly impacts CRC progression.

## Introduction

In tumor development, the influence of the microbiome is profound and multifaceted. Gut bacteria produce metabolites that can induce DNA damage and inflammation, thereby fostering an environment conducive to cancer development [1,2]. Additionally, certain bacterial strains can impair immune surveillance, facilitating the evasion of cancer cells from detection [3]. This impact extends beyond the gastrointestinal tract; the oral and skin microbiomes also influence localized cancers. Furthermore, the microbiome affects therapeutic outcomes, with some bacterial species enhancing drug delivery and immune responses, while others may obstruct these processes [4].

Commonly classified as carcinoma and ranks among the most prevalent and lethal cancers globally in terms of both incidence and mortality [5]. Despite its prominence and severity, research has predominantly concentrated on genetic mutations, environmental factors, and their interactions in CRC development and treatment [4]. Recent advancements, however, underscore the significant role of the microbiota—a complex community of trillions of microorganisms residing within the human body—in modulating these processes [6–8].

Meta-analyses have indicated that the balance between major bacterial phyla, such as Bacillota and Bacteroidota, is crucial for CRC progression. Studies have reported an increase in specific taxa, including Fusobacteriota, *Alistipes*, and Porphyromonadaceae, while taxa such as *Bifidobacterium*, *Lactobacillus*, and *Faecalibacterium* spp. exhibit reduced prevalence [4,9–13]. Notably, these findings often derive from fecal sample analyses, which may not fully represent tumor-microbe interactions. Comparative studies of microbiomes from surrounding healthy tissue, as well as distal and proximal tumor segments, reveal significant variations in bacterial distribution across the colon [14,15].

Considering both consistent and divergent trends in microbiome research, this study seeks to further elucidate the role of the gut microbiome in CRC progression. We analyzed the V3-V4 region of the 16S rRNA gene from colon wall tissue and tumor samples to gain deeper insights into this complex relationship. \.

## Materials and methods

### Samples collection

This study was approved by the ethics committee of the Hermanos Ameijeiras Clinical Surgical Hospital. All participants provided written informed consent for the use of their archived specimens in medical research.

Between January and September 2023, patients who underwent curative surgical resection for colorectal cancer (CRC) were enrolled in the study. All diagnoses were confirmed by pathological analysis, and the expression of mismatch repair (MMR) proteins was assessed using immunohistochemical (IHC) techniques. For microbiome analysis, samples from both the colon wall and the tumor interior were collected by scraping the surgical specimens at the time of surgical intervention.

Samples for microbiome analysis, were collected from consenting patients at the time of surgery and de-identified. Tissue samples were stored in DNA/RNA Shield (Zymo Research, Irvine, CA, USA) and archived. Samples were retrieved on October 12, 2023, for microbiome

analysis. Samples were shipped to EzBiome ((Gaithersburg, MD, USA) for processing, sequencing, and bioinformatics analysis.

## Classification of microsatellite instability based on MMR detection

MMR protein involvement was assessed using an automated immunohistochemistry technique on the Benchmark Ultra system (Roche) according to the manufacturer's instructions. Tissue sections, 2 to 5 nm thick, were prepared from paraffin-embedded blocks using a microtome and mounted on positively charged slides labeled by the machine. The slides were then loaded into the system, which was pre-programmed for the specific antibodies: MLH1, MSH2, MSH6, and PMS2.

The slides were first deparaffinized by heating them to 72˚C, followed by various wash protocols depending on the antibody used. These steps included treatment with the CC1 cell conditioner and heating to 100˚C, followed by incubation as an initial step. A 3% hydrogen peroxide solution was applied to the slides, followed by the specific antibody, and the reaction was visualized using a brown chromogen. After the chromogen application, the slides were rinsed with a blueing reagent, counterstained with hematoxylin, and rinsed again. The slides were then thoroughly washed under running water, dehydrated using a series of ethanol washes (75%, 80%, 90%, and 100%), and finally treated with xylene. The slides were mounted with UKITT mounting medium for microscopic examination.

Glandular tissue without atypia or immune cells present in the sample served as an internal control. If non-tumor tissue was not available in the sample, external controls with normal colonic tissue were used. Nuclear expression of MLH1, MSH2, MSH6, and PMS2 proteins indicated an intact (efficient) DNA mismatch repair (MMR) system. The absence of nuclear expression of any of these markers classified the adenocarcinoma as having deficient MMR. Tumors were categorized based on the level of microsatellite instability as high (MSI-H), low (MSI-L), or stable (MSS), while those with negative expression for at least one marker were classified as MSI-H [16,17].

## Microbiome analysis

**16S metagenomic sequencing.** Metagenomics 16S sequencing was performed by EzBiome (Gaithersburg, MD, USA). DNA concentration was measured using the QuantiFluor dsDNA System on a Quantus Fluorometer (Promega, Madison, WI, USA). The 16S rRNA primers targeting the V3-V4 region of the ribosomal transcript were amplified with primer pairs that included gene-specific sequences and Illumina adapter overhangs. The primer sequences used were: IlluminaF: `CCTACGGGNGGCWGCAG` and IlluminaR: `GACTACHVGGGT ATCTAATCC`

Amplicon PCR was performed to amplify templates out of input DNA samples. Briefly, each 25 μL of polymerase chain reaction (PCR) reaction contains 12.5 ng of sample DNA as input, 12.5 μL 2x KAPA HiFi HotStart ReadyMix (Kapa Biosystems, Wilmington, MA) and 5 μL of 1 μM of each primer. PCR reactions were carried out using the following protocol: an initial denaturation step performed at 95˚C for 3min followed by 25 cycles of denaturation (95˚C, 30 s), annealing (55˚C, 30 s) and extension (72˚C, 30 sec), and a final elongation of 5 min at 72˚C. PCR product was cleaned up from the reaction mix with Mag-Bind RxnPure Plus magnetic beads (Omega Bio-tek, Norcross, GA).

A second index PCR amplification, used to incorporate barcodes and sequencing adapters into the final PCR product, was performed in 25 μL reactions, using the same master mix conditions as described above. Cycling conditions were as follows: 95˚C for 3 minutes, followed

by 8 cycles of 95˚C for 30", 55˚C for 30" and 72˚C for 30". A final, 5 minutes' elongation step was performed at 72˚C.

The libraries were normalized with Mag-Bind® EquiPure Library Normalization Kit (Omega Bio-tek, Norcross, GA) then pooled. The pooled library was checked using an Agilent 2200 TapeStation and sequenced (2 x 300 bp paired-end read setting) on the MiSeq (Illumina, San Diego, CA).

All paired end metagenomic sequences used to generate the data in this study have been deposited and registered with the BioProject database at the National Center for Biotechnology Information with a GenBank BioProjectID PRJNA 1150116. These data have been released and available for download and analysis with the following accession numbers: SAMN43270213, SAMN43270214, SAMN43270215, SAMN43270216, SAMN43270217, SAMN43270218, SAMN43270219, SAMN43270220, SAMN43270221, SAMN43270222, SAMN43270223, SAMN43270224, SAMN43270225, SAMN43270226, SAMN43270227, SAMN43270228, SAMN43270229, SAMN43270230, SAMN43270231, SAMN43270232, SAMN43270233, SAMN43270234, SAMN43270235, SAMN43270236, SAMN43270237, SAMN43270238, SAMN43270239, SAMN43270240, SAMN43270241, SAMN43270242, SAMN43270243, SAMN43270244, SAMN43270245, and SAMN43270246.

**Amplicon taxonomic assignment and functional prediction.** Taxonomic profiling of 16S sequencing data was carried out by directly uploading forward and reverse paired end reads to the EzBioCloud microbiome taxonomy profiling platform (www.ezbiocloud.net) as described elsewhere [18]. Briefly, the cloud application of the EzBioCloud detects and filter out sequences of low quality regarding read length (<80 bp or >2,000 bp) and averaged Q values less than 25. Denoising and extraction of non-redundant reads are carried out using DUDE-Seq software[19]. The UCHIME [20] algorithm was applied against the EzBioCloud 16S chimera-free database to check and remove chimera sequencing. Taxonomic assignment was performed using the USEARCH program to detect and calculate the sequence similarities of the query single-end reads against the EzBioCloud 16S database. Sequencing reads are clustered into operational taxonomic units (OTUs) at 97% sequence similarity using the UPARSE algorithm [21]. Reads from each sample were clustered into many OTUs using the UCLUST [22] tool with the above-noted cutoff values. For the EzBioCloud 16S-based microbiome taxonomic profile (MTP) pipeline, the PICRUSt2 algorithm [23] was used to estimate the functional profiles of the microbiome identified using 16S rRNA sequencing. The raw sequencing reads were computed using the EzBioCloud 16S microbiome pipeline with default parameters and discriminating reads that were encountered in the reference database. The functional abundance profiles of the microbiome were annotated based on bioinformatics analyses, specifically by multiplying the vector of gene counts for each OTU by the abundance of that OTU in each sample, using the KEGG (Kyoto Encyclopedia of Genes and Genomes) [24] orthology and pathway database.

**Amplicon comparative statistical and bioinformatic analyses.** The study utilized the EzBioCloud workflow for subsampling, generating taxonomy plots/tables, and constructing rarefaction curves. Species richness, coverage, and alpha diversity indices were calculated. Microbial richness was assessed through abundance-based coverage estimators (Chao1), and Shannon, Simpson, and Phylogenetic α-diversity indices. Wilcoxon rank-sum tests [25] were applied for diversity estimation across groups. Kruskal-Wallis H test [26] and Effect Size (LEfSe) [27] analysis identified taxonomic and functional enrichment between groups. Taxonomic levels with LEfSe values >2 at p-value < 0.05 were deemed statistically significant. R program (version 3.6.3, R Foundation for Statistical Computing) and packages including 'vegan' v2.5–6, 'ggpubr', and 'ggplot2' v3.3.2 were used for statistical analyses and graphics [28]. Microbiome analysis had a rarefaction depth of 25,000 reads. All p-values were two-

tailed, and significance was set at p < 0.05. Benjamini and Hochberg false discovery rate correction corrected errors in null hypothesis testing for multiple comparisons [29].

**Statistical análisis.**   A Shapiro-Wilk test [30] was initially applied to assess the normal distribution of all pathology and immunohistochemical parameters. Statistical analysis for significance of alpha diversity results were determined by Welch's t-test with the assumption of equal variances are violated [31]. Normally distributed variables underwent log-transformation prior to analyses, and outliers were removed following Boxplot analyses. The Spearman's test was employed to examine correlations between the more abundant phyla and clinical data. The Kruskal-Wallis [32] analysis was employed to evaluate significance for all taxonomic comparisons. Statistical significance was set at a p-value < 0.05.

# Results and discussion

## Metagenomic sequence data

A total of 36 samples were sequenced, including a positive and a negative control. All passed QC. The positive control, as expected yielded 10,718 total sequences while the negative control yielded only 122 sequences. The total number of sequences for remaining 34 samples, representing paired colon and tumor biopsy samples were 757,974 with an average of 21,974.53 ± 3,484.78 with a mean read length of 247.81 ± 0.48. The mean GC content for the entire dataset was 52.66% ± 1.48%. No difference in the GC content between the microbiomes of the tumor tissue (52.77%) and the paired colon tissue (52.56%).

## Sample characterization

Microbiome studies involved the analysis of 34 samples, corresponding to 17 individuals. The demographic distribution of surgery individuals according to age showed an average of 70.18 years with the minimum and maximum of 58 and 86 years respectively. The frequency distribution by sex was very similar, 8 female and 9 males, and the colon affected showed that the left colon tumor were predominant in female (n = 5) and the right colon tumor in males (n = 5), also transverse colon (n = 1).

Surgical fragments from the colon were examined both within the tumor and in the adjacent colon walls. Comprehensive characterization of all individuals was conducted through histological and immunohistochemical assessments for colon cancer, as summarized in Table 1.

## Alpha diversity

The alpha diversity of the microbial communities within biopsy materials was evaluated using a comprehensive panel of metrics encompassing species richness, evenness, composition, and phylogenetic diversity. This panel included Chao1, Shannon Diversity, Simpson Diversity, and Phylogenetic Diversity indices. Good's Coverage of Library [33] was calculated as a quality assurance step in the analysis with a mean of 95.36 ± 0.89%. A summary of the results is provided in Table 2.

Wilcoxon rank-sum tests [25] were used to assess significance between colon and tumor microbiomes.

Good's Coverage of Library [33] was calculated as a quality assurance step in the analysis with a mean of 95.36 ± 0.89%. This level of coverage suggests that the sequencing effort has adequately captured the diversity present in the sample, although it's essential to keep in mind that rare taxa may still be present but not detected [34]. This study highlights a noticeable decrease in the diversity of the tumor microbiome when compared to healthy colonic tissue.

**Table 1. Characteristics of samples according to histology and immunohistochemical parameters.**

| Patient | Age | Sex | Colon Surgery | MMR d (MSI H or Stable) | Infiltration layer | Differentiation | Histology | Lympho-infiltrate | Crohn like | Metastasis |
|---|---|---|---|---|---|---|---|---|---|---|
| 1 | 86 | F | Right | MLH(MSI-H) | muscle | Poorly differentiated ADC | Mucinous papillar | yes | no | no |
| 2 | 63 | F | Right | Stable | serose | Moderately Differentiated ADC | Mucinous | no | yes | no |
| 3 | 80 | M | Right | MLH1 (MSI-H) | subserose | Moderately Differentiated ADC | Mucoproducer | no | no | no |
| 4 | 58 | M | Right | Stable | serose | Moderately Differentiated ADC | Mucoproducer in papillary areas | no | no | Yes (1 ganglion) |
| 5 | 77 | F | Right | Stable | pericolic fat | Moderately Differentiated ADC | Ulcerated | yes | yes | no |
| 6 | 65 | M | Right | Stable | Muscle layer | Moderately Differentiated ADC | Ulcerated | Yes | No | No |
| 7 | 75 | M | Right | MLH1 (MSI-H) | serose | Poorly differentiated ADC | Ulcerated | No | No | No |
| 8 | 73 | M | Right | Stable | Muscle | Well differentiated ADC | Mucinous | Yes | Yes | No |
| 9 | 64 | F | Left | Stable | Muscle | Moderately Differentiated ADC | Vegetative ulcer | No | No | No |
| 10 | 72 | F | Left | Stable | Serose | Moderately Differentiated ADC | Vegetative ulcer and necrosis | No | No | No |
| 11 | 73 | F | Left | Stable | Serose | Moderately Differentiated ADC | Necrotic areas | Yes | No | No |
| 12 | 67 | F | Left | Stable | Serose | Moderately Differentiated ADC | | Yes | No | No |
| 13 | 71 | M | Left | Stable | serose | Moderately Differentiated ADC | Ulcerated | No | No | No |
| 14 | 71 | M | Left | Stable | Muscle | Moderately Differentiated ADC | Ulcerated, papillary pattern, necrotic areas | Yes | No | No |
| 15 | 60 | M | Left | Stable | Muscle | Moderately Differentiated ADC | Ulcerated | No | No | No |
| 16 | 69 | F | Left | Stable | pericolic | Moderately Differentiated ADC | Ulcerated | No | No | No |
| 17 | 69 | M | Transverse | Stable | subserose | Moderately Differentiated ADC | Ulcerated, necrosis areas | Yes | Yes | No |

ADC: Adenocarcinoma.

MMRd: Mismatch Repair deficient.

**Table 2. Alpha diversity analysis results comparing the diversity between colon samples and the corresponding tumor sample.**

| Alpha Diversity Metric | Mean Values | | Statistics | | |
|---|---|---|---|---|---|
| | Colon (C) | Tumor (T) | % C/T change | T-Test P• | Interpretation |
| **Chao1** | 206.54 | 163.56 | 26.3 | P = 0031 | Significant |
| **Phylogenetic Diversity** | 460.35 | 378.41 | 21.7 | P = 0.971 | Not significant |
| **Shannon Index** | 3.82 | 3.25 | 17.5 | P = 0.034 | Significant |
| **Simpson** | 0.064 | 0.119 | 45.9 | P = 0.145 | Not significant |

*Statistical analysis for significance of alpha diversity results were determined by Welch's t-test with the assumption of unequal variances.

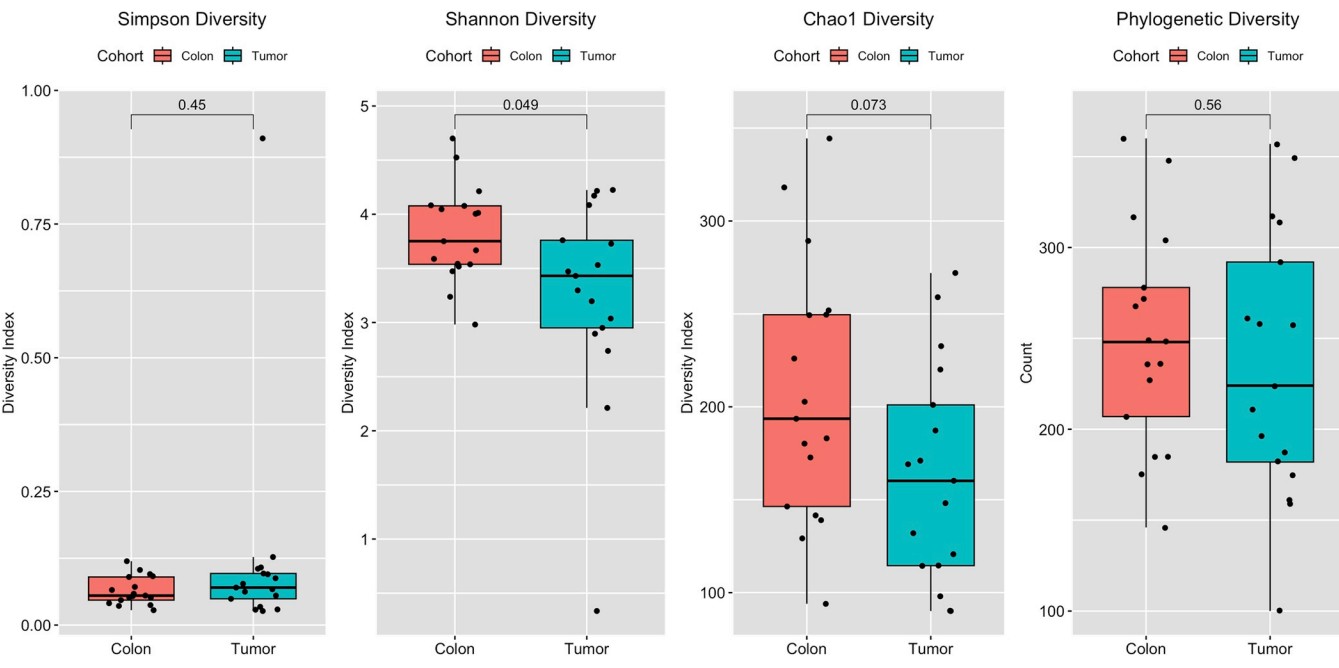

**Fig 1. Alpha diversity results of tumor microbiome compared to healthy colonic tissue.**

The analysis revealed a significant decrease in alpha diversity within the tumor microbiome compared to its corresponding healthy colonic tissue (Table 1). This finding was supported by both Chao1 ($p = 0.031$) and Shannon ($p = 0.034$) indices, indicating a statistically significant reduction in species richness and evenness within the tumor microenvironment. No such changes were noted in Phylogenetic Diversity values. While not statistically significant at the $p < 0.05$ threshold, the Simpson Diversity index displayed lower values in colon tissue compared to tumor tissue, suggesting a potential trend towards reduced species diversity as well as uniformity of species. These findings are visually depicted in Fig 1.

These findings align well with previous studies that have also reported similar observations in CRC. For instance, Ai et al. (2019) [35] reported that the microbial diversity of healthy controls was significantly higher than that of CRC patients, suggesting a significant negative correlation between gut microbiota diversity and CRC stage. Taken together, these results support the idea that both individual microbes and the overall structure of the gut microbiota are co-evolving with CRC.

As for the direction of this co-evolution, it suggests that as CRC progresses, there is a reduction in gut microbiota diversity. This implies that certain microbial populations may be favored or disfavored as the disease advances, potentially impacting the overall balance and composition of the gut microbiome.

The status of the colon walls microbiome can be elucidated by examining the gut microbiome. A decrease in diversity observed within the tumor microbiome likely reflects perturbations in the colon walls microbiome, indicating potential dysbiosis or imbalance associated with CRC development and progression. This suggests that alterations in the gut microbiome could serve as a surrogate marker for assessing the status of the colon walls microbiome, thereby providing valuable insights into the health or disease state of the colon.

Similarly, Murphy et al. [36], Peters et al. ([37], and Cong et al. [38] reported CRC-associated microbiota is characterized by a reduced alpha diversity compared with healthy controls.

Conversely, Li and coworkers [39] reported no significant reduction in the Simpson and Shannon diversity indices between the two cohorts. The results reported in Fig 1 indicate that while there was no significant difference in the Shannon Diversity index (p = 0.092), there was a reduction in the median value of the tumor tissue as compared to the adjacent, normal tissue microbiome. Moreover, this reduction in diversity has been linked to various aspects of tumor progression, including immune evasion, invasion, and metastasis [40].

An aberrant reduction in gut microbiome diversity is a recognized hallmark of tumorigenesis, potentially exerting a pivotal influence on disease initiation and advancement [41–43]. This term implies a departure from the normal microbial composition and function, which can result from various mechanisms. These mechanisms may include dietary factors, lifestyle choices, environmental exposures, genetic predispositions, and medical interventions such as antibiotic usage. While some alterations may be unintentional or incidental, others may indeed be purposeful, driven by factors like dietary changes or medication. Understanding the precise triggers and mechanisms behind these alterations is essential for discerning their controllability. While certain factors leading to microbiome alterations may be modifiable through interventions such as dietary modifications, probiotics, or lifestyle adjustments, others may be less controllable, such as genetic predispositions. Thus, elucidating the controllable aspects of microbiome alterations is crucial for developing targeted interventions to maintain or restore microbial diversity and mitigate the risk of tumorigenesis. The current study further supports the hypothesis that an altered gut microbiome diversity is a hallmark of tumorigenesis, potentially playing a crucial role in disease development and progression.

## Taxonomic composition analysis

The relative abundance (RA) of Bacillota and Bacteroidota appeared similar in both the colon and corresponding tumor tissue, comprising 5.02% and 4.04%, respectively. In contrast, Pseudomonadota emerged as the predominant taxon within the tumor microenvironment, showing a 10.3% higher abundance compared to the colon. Furthermore, Verrucomicrobia was primarily detected in the colon, while Fusobacteriota were prevalent within the tumor microenvironment (Tables 3 and 4).

To explore potential connections between the location of tumors and specific regions of the colon, we conducted an analysis that considered both factors. The findings revealed distinct

**Table 3. High level taxon distribution distribution in the colon and tumor microenvironments.**

| Sample | Colon (%) | Tumor (%) |
|---|---|---|
| Actinobacteriota | 2.5 | 2.6 |
| Bacillota | 54.6 | 50.56 |
| Bacteroidota | 25.42 | 20.4 |
| Cyanobacteria | 0.0247 | 0.0167 |
| Deferribacteres | 0.0008 | 0.0013 |
| Elusimicrobiota | 0.005 | 0.0048 |
| Fusobacteriota | 0.3985 | 1.52 |
| Lentisphaerota | 0.0604 | 0.0273 |
| Mycoplasmatota | 0.2202 | 0.2113 |
| Pseudomonadota | 15.62 | 23.93 |
| Saccharibacteria | 0.0038 | 0.0012 |
| Spirochaetes | 0.0366 | 0.0509 |
| Synergistota | 0.0168 | 0.1369 |
| Verrucomicrobiota | 1.1 | 0.5309 |

**Table 4. Statistical evaluation of phyla distribution between colon and tumor.**

| Phylum | Mean Values | | Statistics | | |
|---|---|---|---|---|---|
| | RA Tumor | RA Colon | % T/C change | T-Test Stat* | Interpretation |
| **Fusobacteriota** | 1.519 | 0.398 | 112.03 | P = 0061 | Significant |
| **Pseudomonadota** | 23.935 | 15.617 | 34.75 | P = 0.0145 | Significant |
| **Verrucomicrobiota** | 0.531 | 1.096 | -106.44 | P = 0048 | Significant |
| **Actinobacteriota** | 2.503 | 2.560 | -3.86 | P = 0.589 | Not Significant |
| **Bacillota** | 54.595 | 50.562 | 7.39 | P = 0.438 | Not Significant |
| **Mycoplasmatota** | 0.211 | 0.220 | -4.20 | P = 0.913 | Not Significant |
| **Bacteroidetes** | 20.931 | 25.838 | -23.44 | P = 494 | Not Significant |

*Statistical analysis for significance of alpha diversity results were determined by Welch's t-test with the assumption of equal variances are violated.

patterns of bacterial distribution associated with tumor location. Bacillota was the most prevalent phylum across all tumor locations, surpassing the abundance of Bacteroidota. Specifically, Fusobacteriota were found in all tumors, with a notable presence of 4.9% specifically in the transverse colon. Pseudomonadota were present across all tumor locations but were notably more abundant in the right and left colon. Actinomycetota, although the least represented phylum in both tumors and the colon overall, showed a relatively higher representation in tumors of the left colon.

Analysis of the ascending (right) colon wall and corresponding tumor revealed an enrichment of Bacteroidota (B) compared to Bacillota (F), evident in significantly higher ($p < 0.001$) B/F ratios in colon ($\bar{X} = 1.066$) as compared to the corresponding tumor (($\bar{X} = 0.4813$). A similar trend was observed in the descending (left) colon (F/B ratios: $\bar{X} = 0.268$ for colon, 0.214 for tumor), albeit of lesser and not significant magnitude that in the right colon. Interestingly, the transverse colon displayed an opposite trend, harboring more Bacteroidota than Bacillota in both tissue types (F/B ratios: 01.031 for colon, add 1.269 for corresponding tumor), suggesting a more balanced bacterial composition in this region. These findings highlight significant regional variations in the gut microbiome composition and potential shifts in the F/B balance associated with tumorigenesis.

We conducted a detailed analysis of Fusobacteriota distribution in healthy colon and corresponding tumor tissues from the same individuals, incorporating MMR classification of the tumors. This analysis revealed notable differences in Fusobacteriota abundance and potentially its association with MMR status. These results are illustrated in Fig 2.

Our study confirms that the four dominant phyla in the intestinal microbiome, Actinomycetota, Bacteroidota, Bacillota, and Pseudomonadota, are also the most abundant phyla in both colon tissue and tumor tissue. This finding is consistent with previous reports on the healthy gut microbiome [45]. Interestingly, we observed a 10.31% increase in the relative abundance of Pseudomonadota in tumor tissue compared to colon tissue.

The role of tumor localization in survival outcomes is debated. Hemminki et al. [46] reported that tumor location played a minor role in survival in a Swedish population, with left-sided tumors having the best prognosis and right-sided and transverse tumors having the worst prognosis. However, a more recent study from Taiwan found that proximal/distal colon cancers had a worse prognosis than rectal cancers [47]. These conflicting findings suggest that the relationship between tumor localization and survival may be complex and influenced by other factors.

The ratio of Bacillota to Bacteroidota (F/B) is a recognized marker of gut dysbiosis, a condition linked to various health problems, including obesity, diabetes, and inflammatory bowel

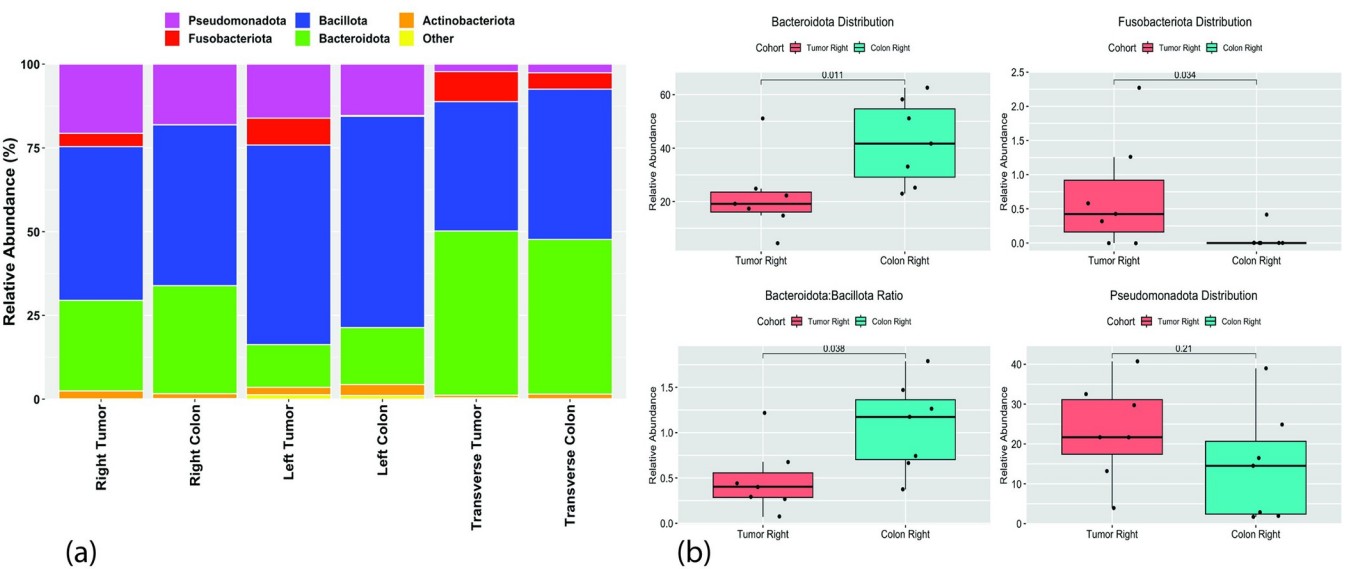

**Fig 2. Phylum distribution along the large intestine.** (a) Bar plot representing the relative abundance of Phyla in the ascending (right), descending (left) and transverse colon. (b) Box plot of representative Phyla in the right column. Statistical significance was determined using the Mann-Whitney U test [44].

disease (IBD) [48–51]. Our study found that the F/B ratio was more imbalanced in the ascending colon as compared to the descending colon. This finding suggests that the ascending colon may be more susceptible to gut dysbiosis, which could potentially contribute to a worse prognosis in this colon segment.

The metabolic pathway of fatty acids exhibits significant activity within both colon and tumor tissues. Studies suggest that Bacillota bacteria demonstrate superior capabilities in fermenting and metabolizing carbohydrates and lipids compared to Bacteroidota. This difference may contribute to the development of inflammatory imbalances[48,52]. Our observation of a more balanced F/B ratio in the transverse colon, associated with a relatively better prognosis, lends support to this hypothesis. Nevertheless, it's essential to acknowledge that other factors likely influence prognosis, and further investigation is necessary to unravel the intricate interplay among gut microbiota composition, metabolic pathways, and colorectal cancer tumor localization.

## Taxonomic biomarkers analysis

Kruskal-Wallis analysis identified 12 taxa with significant differences in relative abundance between tumor and colon samples. These taxa consisted of 3 species, 2 genera, 1 class, and 2 families. To further pinpoint specific taxonomic units with differential abundance, we employed the LEfSe method.

LEfSe analysis revealed Ruminococcaceae family as the most significantly overrepresented taxon in the colon wall, exhibiting a large effect size. Conversely, the *Campylobacteraceae* family showed significant enrichment in the tumor microenvironment. Notably, this family harbored several taxa exclusively associated with the tumor, including Campylobacterales order, *Campylobacter* genus, Epsilon Pseudomonadota class, and *Campylobacter gracilis* species. Interestingly, two additional genera and two species belonging to different families were solely detected in the colon wall (details presented in Table 5).

Analysis of bacterial taxonomy at the genus level demonstrated distinct bacterial profiles between colon and tumor tissues. Specifically, 17 genera showed predominant enrichment in the colon wall. Noteworthy among these were *Fusicatenibacter*, *Lachnospira*, *Akkermansia*,

**Table 5. Taxonomy biomarkers analysis by LEfSe.**

| Taxon Name | Taxon Rank | LDA effect size | p-value | p-value (FDR) | Tumor | Colon |
|---|---|---|---|---|---|---|
| *Ruminococcaceae* | Family | 4.58736 | 0.03561 | 0.80770 | 17.74118 | 26.07059 |
| Campylobacteriales | Order | 3.03889 | 0.04008 | 0.80770 | 0.25294 | 0.05294 |
| *Campylobacter* | Genus | 3.03779 | 0.04008 | 0.80770 | 0.25294 | 0.05294 |
| Epsilon Pseudomonadota | Class | 3.03508 | 0.04008 | 0.80770 | 0.25294 | 0.05294 |
| *Campylobacteraceae* | Family | 3.03408 | 0.04008 | 0.80770 | 0.25294 | 0.05294 |
| AF371735 | Species | 2.18604 | 0.03621 | 0.80770 | 0.00000 | 0.03529 |
| EU794292 | Genus | 2.16622 | 0.03621 | 0.80770 | 0.00000 | 0.03529 |
| PAC001270 | Species | 2.12874 | 0.03621 | 0.80770 | 0.00000 | 0.02941 |
| PAC001270 | Genus | 2.11597 | 0.03621 | 0.80770 | 0.00000 | 0.02941 |
| *Campylobacter gracilis* | Species | 2.08191 | 0.03594 | 0.80770 | 0.02353 | 0.00000 |

PAC001207, *Eubacterium*, and PAC001046, which highly enriched in colon samples (refer to Table 5). Conversely, 14 genera were identified in the tumor microbiome, with 9 genera detected only in the tumor microenvironment (see Table 6 for further details).

### *Fusobacterium* presence and its potential role in colorectal cancer

Our study identified a significantly higher abundance of Fusobacteriota in right-sided tumors with microsatellite instability (MSI) (9.53%) compared to stable tumors (0.66%). This aligns with the known preference of *Fusobacterium* for the ascending colon, which is also linked to MSI and HLM1 hypermethylation [53–58]. However, this presents an apparent contradiction: high relative abundance of *Fusobacterium* suggests poor prognosis, while MSI usually indicates a favorable outcome.

We propose that *Fusobacterium's* role in colon cancer might differ based on tumor side and the underlying mechanism of activation. In tumors with HLM1 deficiency, corresponding to a minority of right-sided cases, *Fusobacterium* interaction with host cells may promote tumor development through increased MSI [58,59], leading to a better prognosis in this specific context. However, most right-sided tumors are stable and have the poorer prognosis. Other microbial interactions and factors likely contribute to this poorer outcome.

Supporting this hypothesis, we observed a moderate *Fusobacterium* presence (5.09%) in left-sided tumors, which generally have a better prognosis [46,47]. Here, *Fusobacterium*-mediated inflammation might contribute to the improved prognosis, like the right-sided MSI scenario.

Transverse colon tumors presented a unique situation: high *Fusobacterium* representation in both tumor (9%) and colon wall (4.9%). The observed equilibrium between Fusobacteriota and Bacteroidota, coupled with the worst prognosis for transverse tumors, suggests distinct inflammatory mechanisms in this region. The high *Fusobacterium* enrichment might be associated with worse clinical outcomes in these metastatic cases [60–62].

It is important to note that our study didn't include metastatic cases. Examining *Fusobacterium* association with malignant transformation at earlier detection stages could provide further insights. Nevertheless, existing evidence suggests *Fusobacterium* as a potential risk factor for CRC development and progression, influencing patient survival [4,13,53,60,63,64].

### Taxonomic biomarkers

Taxon abundance shifts in colon wall and tumor microenvironment suggest microbiome dysbiosis in colorectal cancer. Our study identified distinct family-level dominance in the colon

**Table 6. Bacterial genus taxonomic distribution in colon wall and within tumor.**

| Genus* | Colon† | Tumor† | % Diff |
|---|---|---|---|
| **Actinomyces** | ND | 1.15 | - |
| **Aeromonas** | ND | 1.84 | - |
| **Agathobacter** | 1.48 | 2.05 | -39% |
| Akkermansia | 1.31 | ND | - |
| **Alloprevotella** | 2.1 | 2.16 | -2% |
| Bacteroides | 6.11 | 5.34 | 13% |
| Bifidobacterium | 2.52 | 1.80 | 29% |
| Blautia | 2.12 | 1.14 | 46% |
| **Dialister** | ND | 1.11 | - |
| **Enterococcus** | ND | 1.64 | - |
| **Escherichia** | 5.76 | 9.70 | -67% |
| Eubacterium_g23 | 1.06 | ND | - |
| Faecalibacterium | 5.96 | 2.44 | 59% |
| Fusicatenibacter | 1.57 | ND | - |
| **Fusobacterium** | ND | 1.15 | - |
| Gemella | 1.06 | 3.09 | -192% |
| **Granulicatella** | ND | 1.92 | - |
| Lachnospira | 1.38 | | - |
| **Morganella** | ND | 1.56 | - |
| Oscillibacter | 8.35 | 6.14 | 26% |
| PAC001046_g | 1.03 | | - |
| PAC001115 | 1.76 | 1.53 | 13% |
| PAC001207 | 1.15 | ND | - |
| **PAC001458** | ND | 1.00 | - |
| Prevotella | 14.10 | 10.25 | 23% |
| Pseudoflavonifactor | 1.48 | 1.30 | 12% |
| Ruminicoccus_g2 | 1.99 | 1.42 | 29% |
| Sporobacter | 3.28 | 2.33 | 29% |
| Streptococcus | 2.25 | 4.82 | -114% |
| Sutterella | 1.59 | 1.14 | 28% |
| **Veillonella** | ND | 2.02 | - |

* Genera in **Bold** text represents those genera that are most predominant in the colon wall microbiome and those in normal weight represent the tumor microbiome.

†ND = Not Detected.

wall and tumor microenvironment: *Ruminococcaceae* dominated the colon wall, while *Campylobacteraceae* took precedence in tumors. These findings align with studies showing decreased *Ruminococcaceae* in CRC tumor mucosa compared to healthy individuals [65] and increased abundance of *Campylobacter* in tumors [66,67]. Additionally, we successfully identified *Campylobacter gracilis* specifically in the microbiome of tumor tissue.

Genus- and species-level analysis revealed further differences: *Enterococcus* was exclusively found in tumors, while *Streptococcus* appeared in both tissues with higher abundance in tumors. Notably, *Eubacterium* and several other genera (*Fusicatenibacter*, *Lachnospira*, *Akkermansia*) were solely present in the colon wall. Similarly, Sun et al. [68] reported increased *Enterococcus* and *Streptococcus* and decreased *Clostridium*, *Roseburia*, and *Eubacterium* in colorectal adenoma, partially mirroring our findings.

Our results extend these observations by demonstrating decreased representation of *Bacteroides*, *Lachnospiraceae*, Clostridiales, *and Clostridium* in tumors. This aligns with He and coworkers' report [69] on a comparison of healthy and colon cancer gut microbiomes, where dominant communities in healthy individuals (Clostridiales, *Clostridia*, Bacillota, *Lachnospiraceae*, *Ruminococcaceae*) contrast with tumor-associated communities rich in Pseudomonadota and *Escherichia coli*. Our data indeed detected Campylobacteriota and *Escherichia coli* predominantly in tumors, while Clostridiales, *Clostridium*, Bacillota, *Lachnospiraceae*, and *Ruminococcaceae* resided in the colon wall.

## Functional biomarker identification utilizing PICRUSt2 and MinPath analysis

Functional biomarkers and associated metabolic pathways were identified through a combined approach using PICRUSt2 and MinPath (Minimal set of Pathways) [70]. Kruskal-Wallis H test initially identified 78 ortholog genes exhibiting significant abundance differences between tumor and colon samples. PICRUSt2 analysis was then employed to infer the relative abundance of functional genes from the taxonomic data, while MinPath was subsequently utilized to eliminate potential noise, incomplete data, and pathway redundancy.

PICRUSt2 analysis predicted a total of 10 ortholog genes, while MinPath identified 11 non-redundant ortholog genes with no interference. Notably, three key biomarkers emerged when analyzing the results from both tools: K11076, K10535, and K11329 (Table 5). Further analysis of relative abundance differences revealed that K11076 and K10535 were predominantly enriched in the tumor microenvironment, while K11329 exhibited higher abundance in colon samples (Table 7).

PICRUSt2 analysis predicted four functional metabolic pathways, with necroptosis and fatty acid metabolism exhibiting the highest representation in both tumor and colon microenvironments (Table 6). Notably, MinPath analysis identified a distinct set of three enriched pathways, with no overlap observed with PICRUSt2 predictions. Interestingly, spliceosome pathway emerged as the most prominent in the tumor microbiome, while D-alanine metabolism was most representative in the colon (Table 8).

Functional biomarkers that connect orthologous genes and metabolic pathways offer valuable insights into the contribution of bacterial dysbiosis to colon cancer. Notably, a significant correlation is observed with the K10535 ortholog gene, which is involved in nitrification and the conversion of ammonia to nitrite. The principal mechanism implicated in human cancer likely revolves around the formation of endogenous N-nitroso compounds (NOCs). Estimates indicate that approximately 45% to 75% of human exposure to N-nitroso compounds stems from in vivo processes, although this varies [71]. Intriguingly, our study reveals a higher representation of the K10535 gene within the tumor. Various pathways could be associated with this, including the nitrification process and nitrite production from ammonia sources. One such source involves the conversion of nitrite amino acids to amines through bacterial

**Table 7. Principal functional biomarkers identified by Both PICRUSt and MinPath analysis in colon and tumor microenvironment.**

| Functional Biomarkers | | | Relative Abundance of Genes (MinPath-PICRUSt) | | |
|---|---|---|---|---|---|
| Ortholog | Definition | p-value PICRUSt | p-value MinPath | Colon | Tumor |
| K11076 | Putrescine transport system | 0.04391024 | 0.0371753 | 11.4 | 13.58 |
| K10535 | Nitrification, ammonia to nitrite | 0.0154208 | 0.01382686 | 0.0053 | 0.028 |
| K11329 | SasA-RpaAB (circadian timing mediating) two-component regulatory system | 0.02007525 | 0.01517109 | 6.47 | 4.41 |

**Table 8. Principal functional pathways identified by Either PICRUSt or MinPath in colon and tumor microenvironment.**

| Pathway (PICRUSt) | Definition | p-value PICRUSt | Relative Abundance | |
| --- | --- | --- | --- | --- |
| | | | Colon | Tumor |
| ko04016 | MAPK signaling pathway—plant | 0.0371753 | 5.75 | 5.08 |
| ko04217 | Necroptosis | 0.02406742 | 14.71 | 15.3 |
| ko01212 | Fatty acid metabolism | 0.04391024 | 41.67 | 40.11 |
| ko00430 | Taurine and hypotaurine metabolism | 0.043910242 | 0.085 | 0.091 |
| **Pathway (MinPath)** | **Definition** | **p-value MinPath** | **Relatice Abundance** | |
| | | | Colon | Tumor |
| ko04260 | Cardiac muscle contraction | 0.0293964 | 0.025 | 0.077 |
| ko03040 | Spliceosome | 0.04391024 | 38.87 | 46.87 |
| ko00473 | D-Alanine metabolism | 0.02406742 | 45.09 | 35.16 |
| ko04141 | Protein processing in endoplasmic reticulum | 0.047647027 | 0.98 | 0.84 |

decarboxylation, followed by N-nitrosation in the presence of nitrite as a nitrosating agent, resulting in the generation of N-nitroso compounds (NOCs). These NOCs are highly mutagenic, and additional exogenous sources include processed, burned, or cured meats [45].

Additional source of nitrite are plants. Plants nitrate and ammonium transporters are responsible for nitrate and ammonium translocation from the soil into the roots. After absorption the nitrogen metabolism pathway incorporates the nitrogen into organic compounds via glutamine synthetase [72]. The excessive use of nitrate fertilizers can lead to health and environmental issues, including CRC [73,74].

Also of significance is the ortholog K11329, which corresponds to the SasA-RpaAB two-component regulatory system of circadian timing. Circadian rhythms in the intestines are regulated by a complex interplay of signals from the central circadian clock, feeding/fasting cycles, gut microbiota, and hormonal signaling, optimizing various intestinal functions to occur at appropriate times throughout the day [75,76]. CRC is one of the cancers closely associated with circadian disruption [77]. The loss of circadian rhythms in the intestine leads to aberrant regulation of stem cell signaling pathways and increased tumor initiation [78].

Moreover, the ortholog K11076, identified as the putrescine transport system, exhibits higher expression levels within the tumor. Putrescine, an organic cation serving as a precursor for elevated polyamine biosynthesis, holds significance in various biological processes such as cell proliferation, differentiation, and chromatin remodeling. The increased polyamine biosynthesis in neoplastic cells has been documented [79,80]. Certain bacterial members of the microbiome convert amino acids, particularly arginine and ornithine, to polyamines or indole derivatives upon reaching the colon. Luminal polyamines in the small intestine are primarily sourced from diets, whereas gut microbiota regulate luminal polyamine concentrations in the colon [81]. Notably, polyamine concentrations demonstrate an increase in CRC tissues compared to healthy tissues [81], establishing them as potential biomarkers for occurrence and progression in various tumors, including colorectal cancer [82,83].

A diet rich in certain amino acids, notably arginine and ornithine, can promote the production of polyamines and indole derivatives by specific bacterial members of the microbiome upon reaching the colon. Luminal polyamines in the small intestine primarily originate from dietary sources, while the gut microbiota regulate polyamine concentrations in the colon [81]. Elevated polyamine concentrations have been observed in CRC tissues compared to healthy tissues, suggesting their potential as biomarkers for tumor occurrence and progression, including colorectal cancer [81].

Our analysis identified six potentially relevant metabolic pathways associated with tumor formation. Notably, MinPath revealed a 10% increase in D-alanine metabolism within the colon wall compared to the tumor itself. Conversely, the spliceosome pathway was more active in the tumor.

Further insights came from PICRUSt predictions, which highlighted several microbiome-activated pathways with potential roles in tumorigenesis. These included the MAPK signaling pathway–plant, Necroptosis, Fatty acid metabolism, and protein processing in the endoplasmic reticulum.

**D-alanine metabolism.** D-alanine, a crucial component of bacterial cell walls, is noteworthy as mammals lack the ability to synthesize it internally. Instead, they depend on their intestinal microbiota for its production, essential for the structure of bacterial peptidoglycan. Some D-alanine is absorbed and circulated through the bloodstream before being excreted in urine [84]. Although the precise mechanisms governing D-alanine uptake, transport, and metabolism remain largely unknown, recent studies have revealed intriguing circadian fluctuations in both rodents and humans [85]. These findings align with our observations of circadian-related gene induction in colon cancer.

Our results suggest that elevated D-alanine biosynthesis gene levels in tumor biopsies could be a potential predictor biomarker and therapeutic target for colon cancer. However, further research is crucial to confirm this hypothesis and elucidate the precise role of D-alanine in the disease process.

**Spliceosome.** Alternative splicing, a key cellular process generating diverse proteins from single genes, is increasingly linked to colon cancer development and prognosis [86]. Notably, it might affect the expression of HLM1, promoting tumor cell growth through microsatellite instability (MSI) [87]. Interestingly, our findings resonate with this concept. We observed a predominant presence of Fusobacteriota Phylum (9.53%) in the right tumors with MMR deficiency (MMRd), which lacked HLM1 expression. This suggests that *Fusobacterium* might contribute to aberrant alternative splicing, leading to dysfunctional proteins with altered functional domains, a characteristic of MSI-high tumors (MSI-H).

**Mitogen-activated protein kinases (MAPK) signaling pathway.** MAPKs, a family of 14 protein kinases, play a crucial role in various cellular processes like gene expression, cell growth, and death [88]. Mutations in one specific MAPK, BRAF, have been linked to the development of colorectal cancer. These mutations lead to the constant activation of BRAF, ultimately triggering uncontrolled cell growth through the MAPK pathway. Interestingly, BRAF mutations, particularly the V600E variant, are often associated with microsatellite instability (MSI) in advanced colorectal cancer [89]. This aligns with previous findings [90] and supports the potential role of MAPK pathway dysregulation in colon cancer development.

**Fatty acid metabolism.** Polyunsaturated fatty acids (PUFAs) are known to influence the epigenome, including DNA methylation, in colorectal cancer. However, the exact mechanisms by which PUFAs impact these epigenetic changes and gene expression in both healthy and cancerous human cells remain unclear [91]. Our findings suggest that bacterial fatty acid metabolism within the tumor may play a role in activating epigenetic modifications, potentially contributing to tumorigenesis. This aligns with previous observations by Allali and co-authors [90]. These insights could be valuable for patient education regarding the potential impact of diet on tumor growth and for making specific dietary recommendations.

**Necroptosis.** Necroptosis, a programmed cell death pathway triggered by the tumor necrosis factor (TNF) family, plays a role in eliminating damaged cells. It involves key proteins like RIPK1, RIPK3, and MLKL to dismantle them from within [92]. Interestingly, our findings resonate with a recent study by Zhang et al. [93], which identified a necroptosis-related gene signature predicting prognosis in colon adenocarcinoma. Our research suggests that the

microbiome may initiate necroptosis in cancer cells, although the specific bacterial species or metabolites responsible and the precise mechanism of this activation remain unclear. Further investigation is required to elucidate these intriguing connections. It is noteworthy that a healthy microbiota, along with metabiotics and postbiotics derived from probiotics, could potentially induce necroptosis in cancer cells.

**Taurine and hypotaurine metabolism.** Taurine (2-aminoethylsulfonic acid) is the most abundant free amino acid in the body, and is mainly derived from the diet, but can also be produced endogenously from cysteine The synthesis of taurine through the cysteine sulfinic acid or transsulfuration pathway, produces hypotaurine by hypotaurine dioxygenase [94,95]. Taurine is a crucial molecule used to conjugate bile acids (BAs) in the liver. In the gastrointestinal tract, BAs are deconjugated by enteric bacteria, resulting in high levels of unconjugated BAs and free taurine. The free form of taurine has been shown to have anti-inflammatory properties [96] and has been associated with inhibited growth of harmful bacteria, including Pseudomonadota [97], and also increasing the production of SCFA [98] thus, impacting the environment of resident gut bacteria. The enriching of gastrointestinal tract with BAs-tolerant taxa such as the families of *Ruminococcaceae* and *Lachnospiraceae*, the genera *Bacteroides* and *Bilophila*, as well as several other taxa, including *Campylobacter*, *Salmonella*, and *E. coli*, have been associated with host inflammation.[94] Our results are in concordance with this study. We found that the taurine and hypotaurine metabolism are represented less than 1% and it could be explained because the taxonomy both in colon wall and intra tumor microenvironment showed the prevalence of BA-tolerant taxa.

**Protein processing in endoplasmic reticulum.** The processing of protein in endoplasmic reticulum (ER) is another pathway represented at less than 1%. The low expresion of protein in ER could be explained by two connected factors: the endoplasmic reticulum stress (ER stress) [99] and the accumulation of misfolded proteins [100]. To survive, cancer cells are subjected to various internal and external adverse factors that result in the accumulation of unfolded proteins in the endoplasmic reticulum, which leads to a condition termed endoplasmic reticulum (ER) stress and triggers the unfolded protein response (UPR). Both aspect in colon cancer could be connected to the gut bacterial metabolites. An *in vitro* study identified some gut microbial metabolites that modulate the ER stress pathway.However, the wide spectrum of bacterial metabolites that interact with ER stress signaling, as well as mechanisms by which these molecules impact this pathway, stay undefined [101].

## Conclusions

This study offers a unique and comprehensive perspective on the role of the microbiome in CRC development and progression. By examining both the colon wall and tumor tissues, the study provides a holistic understanding of microbial influences on CRC. Through analyses of alpha diversity, taxonomic composition, functional biomarkers, and metabolic pathways, the study illustrates the complex interplay between gut microbes and CRC. Notably, the investigation into *Fusobacterium*'s differential abundance based on tumor side and microsatellite instability adds depth to our understanding of CRC prognosis. Furthermore, the identification of potential biomarkers and therapeutic targets related to dysregulated metabolic pathways in CRC highlights avenues for personalized treatment strategies. Overall, this study underscores the importance of embracing the complexity of the microbiome in cancer research and the development of targeted interventions for CRC patients. Further research building upon these findings will be instrumental in advancing our understanding of CRC pathogenesis and improving patient outcomes.

## Acknowledgments

We extend our sincere gratitude to Stellar Biotics for their generous donation of the reagents essential for this study. We also deeply appreciate the invaluable assistance provided by Dr. Nur Hasan of EzBiome, along with the expedited microbiome analysis. Our heartfelt thanks go to Dr. Miguel H. Estévez del Toro, Director of the Hermanos Ameijeiras Hospital and Eduardo López, Eng., for their unwavering support in initiating and conducting the study.

## Author Contributions

**Conceptualization:** Liubov Sichel, Ernesto Arteaga, Raúl De Jesus Cano.

**Data curation:** Gissel García Menéndez, Vilma Fleites.

**Formal analysis:** Gissel García Menéndez, Raúl De Jesus Cano.

**Funding acquisition:** Liubov Sichel.

**Investigation:** Maria del Consuelo López, Yasel Hernández, Ernesto Arteaga, Marisol Rodríguez, Vilma Fleites, Lipsy Teresa Fernández.

**Methodology:** Liubov Sichel, Maria del Consuelo López, Yasel Hernández, Ernesto Arteaga, Marisol Rodríguez, Vilma Fleites, Lipsy Teresa Fernández.

**Project administration:** Gissel García Menéndez, Raúl De Jesus Cano.

**Resources:** Liubov Sichel, Yasel Hernández, Ernesto Arteaga, Marisol Rodríguez, Vilma Fleites, Lipsy Teresa Fernández, Raúl De Jesus Cano.

**Supervision:** Gissel García Menéndez, Ernesto Arteaga, Raúl De Jesus Cano.

**Validation:** Maria del Consuelo López, Raúl De Jesus Cano.

**Writing – original draft:** Gissel García Menéndez.

**Writing – review & editing:** Liubov Sichel, Raúl De Jesus Cano.

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
