## [Decision Letter · Decision Letter 0]

13 Aug 2024

PONE-D-24-11510From Colon Wall to Tumor Niche: Unraveling the Microbiome's Role in Colorectal Cancer ProgressionPLOS ONE

Dear Dr. Cano,

Thank you for submitting your manuscript to PLOS ONE. After careful consideration, we feel that it has merit but does not fully meet PLOS ONE’s publication criteria as it currently stands. Therefore, we invite you to submit a revised version of the manuscript that addresses the points raised during the review process.

We look forward to receiving your revised manuscript.

Kind regards,

MUHAMMAD SHAHID RIAZ RAJOKA, Ph.D.

Academic Editor

PLOS ONE

Reviewers' comments:

Reviewer's Responses to Questions

**Comments to the Author**

1. Is the manuscript technically sound, and do the data support the conclusions?

Reviewer #1: Yes

Reviewer #2: Yes

2. Has the statistical analysis been performed appropriately and rigorously? 

Reviewer #1: Yes

Reviewer #2: Yes

3. Have the authors made all data underlying the findings in their manuscript fully available?

Reviewer #1: Yes

Reviewer #2: Yes

4. Is the manuscript presented in an intelligible fashion and written in standard English?

Reviewer #1: Yes

Reviewer #2: Yes

5. Review Comments to the Author

Reviewer #1: The manuscript is well structured. It is completing all the aspects accordingly. There is a significant data showing less diverse bacterial environment in the cancer compromised colon as compared to healthy colon. there is some interesting data showing Well differentiated Adenocarcinoma with Crohn like appearance, Microsatellite instability in aged Male patients. Moreover the data supports the CRC progression with the dysbiosis.

Reviewer #2: Please formalize the introduction section. Additionally, expand the methods and results sections to enhance their comprehensiveness, ensuring all critical details and explanations are thoroughly covered.

6. PLOS authors have the option to publish the peer review history of their article (what does this mean?). If published, this will include your full peer review and any attached files.

Reviewer #1: **Yes: **Hammad Ali Hassan

Reviewer #2: No

---

## [Author Response · Author response to Decision Letter 0]

21 Aug 2024

a. Reviewer #1: We appreciate the positive feedback and have ensured that all the data available from the study are included in the body of the manuscript. Additionally, we have deposited all the SRA paired end reads, from which all the microbiome analyses were based, in the NCBI SRA database with the BioProject ID number PRJNA 1150116. 

b. Reviewer #2: We have formalized the introduction section as suggested. Additionally, we expanded the Methods and Results sections to improve comprehensiveness, ensuring that all critical details and explanations are thoroughly covered. Specifically, we have revised the descriptions of 16S metagenomic sequencing under Microbiome Analysis and the Classification of microsatellite instability based on MMR detection to include more detailed explanations

---

## [Editor Report · Decision Letter 1]

17 Sep 2024

From Colon Wall to Tumor Niche: Unraveling the Microbiome's Role in Colorectal Cancer Progression

PONE-D-24-11510R1

Dear Dr. Raul,

We’re pleased to inform you that your manuscript has been judged scientifically suitable for publication and will be formally accepted for publication once it meets all outstanding technical requirements.

Kind regards,

MUHAMMAD SHAHID RIAZ RAJOKA, Ph.D.

Academic Editor

PLOS ONE
---

## [Editor Report · Acceptance letter]

9 Oct 2024

PONE-D-24-11510R1 

PLOS ONE

Dear Dr. Cano, 

I'm pleased to inform you that your manuscript has been deemed suitable for publication in PLOS ONE. Congratulations! Your manuscript is now being handed over to our production team.

Kind regards, 

on behalf of

Dr. MUHAMMAD SHAHID RIAZ RAJOKA 

Academic Editor

PLOS ONE